# Novel Antiviral Activities of Obatoclax, Emetine, Niclosamide, Brequinar, and Homoharringtonine

**DOI:** 10.3390/v11100964

**Published:** 2019-10-18

**Authors:** Petter I. Andersen, Klara Krpina, Aleksandr Ianevski, Nastassia Shtaida, Eunji Jo, Jaewon Yang, Sandra Koit, Tanel Tenson, Veijo Hukkanen, Marit W. Anthonsen, Magnar Bjoras, Magnus Evander, Marc P. Windisch, Eva Zusinaite, Denis E. Kainov

**Affiliations:** 1Department of Clinical and Molecular Medicine, Norwegian University of Science and Technology, 7028 Trondheim, Norway; petteria@stud.ntnu.no (P.I.A.); klarak@stud.ntnu.no (K.K.); aleksandr.ianevski@ntnu.no (A.I.); marit.w.anthonsen@ntnu.no (M.W.A.); magnar.bjoras@ntnu.no (M.B.); 2Institute of Technology, University of Tartu, Tartu 50090, Estonia; nastassia.shtaida@ut.ee (N.S.); sandra.koit@ut.ee (S.K.); tanel.tenson@ut.ee (T.T.); eva.zusinaite@ut.ee (E.Z.); 3Applied Molecular Virology Laboratory, Institut Pasteur Korea, 696 Sampyung-dong, Bundang-gu, Seongnam-si, Gyeonggi-do 13488, Korea; eunji.jo@ip-korea.org (E.J.); jaewon.yang@ip-korea.org (J.Y.); marc.windisch@ip-korea.org (M.P.W.); 4Institute of Biomedicine, University of Turku, 20520 Turku, Finland; veijo.hukkanen@utu.fi; 5Department of Clinical Microbiology, Virology, Umeå University, 90185 Umeå, Sweden; magnus.evander@umu.se

**Keywords:** virus, broad-spectrum antiviral, antiviral agent, drug target, systems biology

## Abstract

Viruses are the major causes of acute and chronic infectious diseases in the world. According to the World Health Organization, there is an urgent need for better control of viral diseases. Repurposing existing antiviral agents from one viral disease to another could play a pivotal role in this process. Here, we identified novel activities of obatoclax and emetine against herpes simplex virus type 2 (HSV-2), echovirus 1 (EV1), human metapneumovirus (HMPV) and Rift Valley fever virus (RVFV) in cell cultures. Moreover, we demonstrated novel activities of emetine against influenza A virus (FLUAV), niclosamide against HSV-2, brequinar against human immunodeficiency virus 1 (HIV-1), and homoharringtonine against EV1. Our findings may expand the spectrum of indications of these safe-in-man agents and reinforce the arsenal of available antiviral therapeutics pending the results of further in vitro and in vivo tests.

## 1. Introduction

Every year, emerging and re-emerging viruses, such as Ebola virus (EBOV), Marburg virus (MARV), and Rift Valley fever virus (RVFV), surface from natural reservoirs and kill people [1,2]. In addition, influenza A virus (FLUAV), human immunodeficiency (HIV-1), herpes simplex (HSV), and other viruses regularly infect human population and represent substantial public health and economic burden [3,4]. The World Health Organization (WHO) and the United Nations (UN) have called for better control of viral diseases (https://www.who.int/blueprint/priority-diseases/en/; https://sustainabledevelopment.un.org/). Developing novel virus-specific vaccines and antiviral drugs can be time-consuming and costly [5,6]. In order to overcome these time and cost issues, academic institutions and pharmaceutical companies have focused on the repositioning of existing antivirals from one viral disease to another, considering that many viruses utilize the same host factors and pathways to replicate inside a cell [6,7,8,9,10,11,12,13,14,15].

Broad-spectrum antiviral agents (BSAAs) are small-molecules that inhibit a wide range of human viruses. We have recently reviewed approved, investigational and experimental antiviral compounds and identified 108 BSAAs, whose pharmacokinetics (PK) and toxicity had been studied in clinical trials [16]. We tested 40 of these BSAAs against human metapneumovirus (HMPV), hepatitis C virus (HCV), cytomegalovirus (CMV), and hepatitis B virus (HBV). We demonstrated novel antiviral effects of azacytidine, itraconazole, lopinavir, nitazoxanide, and oritavancin against HMPV, as well as cidofovir, dibucaine, azithromycin, gefitinib, minocycline, oritavancin, and pirlindole against HCV [17]. We also tested 55 BSAAs, including these 40, against FLUAV, RVFV, echovirus 1 (EV1), ZIKV, CHIKV, RRV, HIV-1 and HSV-1. We identified novel activities for dalbavancin against EV1, ezetimibe against HIV-1 and ZIKV, and azacytidine, cyclosporine, minocycline, oritavancin and ritonavir against RVFV [18].

Here, we evaluated the efficacy of 43 BSAAs, which do not overlap with 55 agents we tested before. We identified novel in vitro activities of obatoclax and emetine against HSV-2, EV1, HMPV and RVFV. Moreover, we demonstrated novel antiviral effects of emetine against FLUAV, niclosamide against HSV-2, brequinar against HIV-1, and homoharringtonine against EV1 in vitro. 

## 2. Materials and Methods

### 2.1. Compounds

To identify potential BSAAs, we have reviewed approved and investigational safe-in-man antiviral agents using drug bank and clinical trials websites, respectively. In addition, we reviewed investigational and approved safe-in-man antibacterial, antifungal, antiprotozoal, antiemetic, etc. agents, for which antiviral activities have been reported in PubMed. By excluding vaccines and interferons, we identified 108 molecules that inhibit the replication of several viruses in man [16]. Most recently, novel antiviral activities have been reported for some of these agents [17]. Forty-three compounds were used in this study, and their suppliers and catalogue numbers are summarized in Appendix A. To obtain 10 mM stock solutions compounds were dissolved in dimethyl sulfoxide (DMSO, Sigma-Aldrich, Steinheim, Germany) or milli-Q water. The solutions were stored at −80 °C until use.

### 2.2. Cells

Madin–Darby canine kidney (MDCK, American Type Culture Collection (ATCC)) and African green monkey kidney epithelial (Vero-E6, ATCC) cells were grown in Dulbecco’s Modified Eagle’s medium (DMEM; Gibco, Paisley, Scotland) supplemented with 100 U/mL penicillin and 100 μg/ml streptomycin mixture (Pen/Strep; Lonza, Cologne, Germany), 2 mM l-glutamine, and 10% heat-inactivated fetal bovine serum (FBS; Lonza, Cologne, Germany). Human telomerase reverse transcriptase-immortalized retinal pigment epithelial (RPE, ATCC) cells were grown in DMEM-F12 medium supplemented with Pen/Strep, 2 mM l-glutamine, 10% FBS, and 0.25% sodium bicarbonate (Sigma-Aldrich, St. Louis, USA). ACH-2 cells, which possess a single integrated copy of the provirus HIV-1 strain LAI (NIH AIDS Reagent Program), were grown in RPMI-1640 medium supplemented with 10% FBS and Pen/Strep. TZM-bl cells were grown in DMEM supplemented with 10% FBS and Pen/Strep. Human lung adenocarcinoma epithelial A549 cells were cultured in DMEM medium containing 10% FBS and Pen/Strep. A549-Npro cells (kindly provided by Prof. Steve Goodbourn, University of London), which stably express BVDV Npro protein, which inhibits IFN production, were cultured in DMEM containing 10% FBS, Pen/Strep, and 10 μg/mL puromycin. All cell lines were grown in a humidified incubator at 37 °C in the presence of 5% CO2.

### 2.3. Viruses

HSV-2 strain G was from the ATCC. EV1 (Farouk strain; ATCC) was from Prof. Marjomäki (University of of Jyväskylä) [19]. RVFV encoding the far-red fluorescent protein instead of non-structural (NS) protein (RVFV-RFP) was from Profs. Hartmut Hengel and Friedemann Weber (University Medical Center Freiburg) [20]. HMPV NL/1/00 strain, encoding green fluorescent protein (HMPV-GFP), was from ViroNovative and Erasmus MC [21]. The GFP-expressing influenza A/PR/8-NS116-GFP strain (FLUAV-GFP) was generated by Drs. Andrej Egorov (Vienna) [22]. 

All the experiments with viruses were performed in compliance with the guidelines of the national authorities using appropriate biosafety laboratories under appropriate ethical and safety approvals. FLUAV-GFP was amplified in a monolayer of MDCK cells in DMEM containing Pen/Strep, 0.2% bovine serum albumin, 2 mM l-glutamine, and 1 μg/mL l-1-tosylamido-2-phenylethyl chloromethyl ketone-trypsin (TPCK)-trypsin (Sigma-Aldrich, St. Louis, USA). HMPV-GFP, RVFV-RFP and the wild-type HSV-2 strain were amplified in a monolayer of Vero-E6 cells in the DMEM medium containing Pen/Strep, 0.2% bovine serum albumin, 2 mM l-glutamine, and 1 μg/mL TPCK-trypsin. EV1 was amplified in a monolayer of A549 cells in the DMEM media containing Pen/Strep, 0.2% bovine serum albumin, and 2 mM l-glutamine.

For the production of HIV-1, 6 × 10^6^ ACH-2 cells were seeded in 10 mL medium. Virus production was induced by the addition of 100 nM phorbol-12-myristate-13-acetate. The cells were incubated for 48 h. The HIV-1-containing medium was collected. The HIV-1 concentration was estimated by measuring the concentration of HIV-1 p24 in the medium using anti-p24-ELISA, which was developed in-house. Recombinant purified p24 protein was used as reference. The virus stocks were stored at −80 °C. 

### 2.4. Microscopy

Approximately 4 × 10^4^ RPE cells were seeded per well in 96-well plates. The cells were grown for 24 h in DMEM-F12 medium supplemented with 10% FBS, and Pen/Strep. The medium was replaced with DMEM-F12 medium containing 0.2% bovine serum albumin, 2 mM l-glutamine, and 1 μg/mL TPSK-trypsin. The compounds were added to the cells in 3-fold dilutions at seven different concentrations starting from 10 or 30 μM. Saliphenylhalamide, ABT-263 and DMSO were added to the control wells. Saliphenylhalamide is an inhibitor of cellular vacuolar ATPase, which protects cells from virus-mediated death, whereas ABT-263 is an inhibitor of anti-apoptotic Bcl-2 proteins, which facilitates death of cells with viral nucleic acids [23,24,25,26,27,28]. RPE cells were infected with HSV-2, FLUAV-GFP, HMPV-GFP or RVFV-RFP viruses at multiplicity of infections (moi) of 0.1, 0.5, 0.1 and 1, respectively. HSV-2-infected RPE cells were imaged after 72 h in the phase-contrast mode. RVFV-mediated RFP expression and FLUAV-mediated GFP expression were visualized after 24 h, whereas HMPV-mediated GFP expression was recorded after 96 h using fluorescent microscopy (Zeiss Observer Z1, Zaventem, Belgium). Image J software (v.IJ 1.46r, NIH) was used to determine fluorescent intensities.

### 2.5. Cell Viability and Toxicity Assays

RPE cells were treated with BSAAs or control compounds as described above and infected with HSV-2, EV1, FLUAV-GFP, HMPV-GFP or RVFV-RFP viruses at multiplicity of infections (moi) of 0.1, 0.1, 0.5, 0.1 and 1, respectively. After 48 h of infection, the medium was removed from the cells. The viability of mock- and virus-infected cells were measured using Cell Titer Glow assay (CTG; Promega, Madison, USA). The luminescence/fluorescence were read with a PHERAstar FS plate reader (BMG Labtech, Ortenberg, Germany). 

For testing compound toxicity and efficacy against HIV-1, approximately 4 × 10^4^ TZM-bl cells were seeded in each well of a 96-well plate. TZM-bl cells express firefly luciferase under control of HIV-1 long terminal repeat (LTR) promoter allowing quantitation of the viral infection (tat-protein expression by integrated HIV-1 provirus) using firefly luciferase assay. The cells were grown for 24 h in cell growth medium. Compounds were added to the cells in three-fold dilutions at seven different concentrations starting from 30 μM. No compounds were added to the control wells. The cells were infected with HIV-1 (corresponding to 300 ng/mL of HIV-1 p24) or mock. At 48 hpi, the media was removed from the cells, the cells were lysed, and firefly luciferase activity was measured using the Luciferase Assay System (Promega, Madison, WI, USA) and PHERAstar FS plate reader. In a parallel experiment, Cell Tox Green reagent (CTxG; Promega, Madison, WI, USA) was added to the cells and fluorescence was measured with a plate reader.

The half-maximal cytotoxic concentration (CC_50_) for each compound was calculated based on viability/death curves obtained on mock-infected cells after non-linear regression analysis with a variable slope using GraphPad Prism software version 7.0a. The half-maximal effective concentrations (EC_50_) were calculated based on the analysis of reporter protein expression or the viability/death of infected cells by fitting drug dose–response curves using four-parameter (*4PL*) logistic function *f*(*x*):
$$f\left(x\right)=A_{min}+\frac{A_{max}-A_{min}}{1+{\left(\frac{x}{m}\right)}^{\lambda }},$$
where *f*(*x*) is a response value at dose *x*, *A_min_* and *A_max_* are the upper and lower asymptotes (minimal and maximal drug effects), *m* is the dose that produces the half-maximal effect (EC_50_ or CC_50_), and *λ* is the steepness (slope) of the curve. A relative effectiveness of the drug was defined as selectivity index (*SI* = CC_50_/EC_50_). The threshold of SI used to differentiate between active and inactive compounds was 3.

### 2.6. Drug Combination Experiment

RPE cells were treated with combinations of increasing concentrations of obatoclax and emetine. The cells were infected with FLUAV-GFP at moi 0.5. After 24 h, GFP fluorescence was recorded using fluorescent microscopy. In a parallel experiment, the viability of infected cells was measured using the CTG assay. To test whether the drug combinations act synergistically, the observed responses were compared with expected combination responses. The expected response of the emetine-obatoclax drug combination on the viability of FLUAV- and mock-infected RPE cells was calculated using Bliss reference model [29]. 

### 2.7. Virus Titration

For testing the production of HSV-2 and EV1 viruses in compound-treated and non-treated RPE cells, the media from the cells were serially (10-fold) diluted, starting from 10^−3^ to 10^−8^ in serum-free growth media containing 0.2% bovine serum albumin, and applied to a monolayer of A549-Npro cells in 12-well plates. After one hour, cells were overlaid with growth medium containing 1% carboxymethyl cellulose and 1% FBS and incubated for 72 h. The cells were fixed and stained with crystal violet dye. The plaques were calculated in each well. The titers were expressed as plaque-forming units per mL (PFU/mL).

## 3. Results

### 3.1. Forty-Three BSAAs Target 52 Viruses

Forty-three safe-in-man BSAAs used in this study reached different stages of drug development process (Figure 1, Appendix A). Altogether, these BSAAs inhibit the replication of 52 viruses belonging to (−) single-stranded (ss)RNA, (+)ssRNA, ssRNA-reverse transcriptase (RT), ssDNA, double-stranded (ds)DNA, or dsDNA-RT virus groups.

### 3.2. Obatoclax, Emetine, Niclosamide and Ganciclovir Inhibit HSV-2 Replication in RPE Cells

We tested 43 BSAAs against wild-type HSV-2 in RPE cells. Seven different concentrations of the compounds were added to virus- or mock-infected cells. Cell viability was monitored by microscopy and the CTG assay. After the initial screening, we identified four compounds (obatoclax, emetine, niclosamide and ganciclovir) that at none-cytotoxic concentrations rescued cells from virus-mediated death (Figure 2A). To determine the efficiency and toxicity of these HSV-2 inhibitors, we measured the viability of mock- and virus-infected cells after 72 h using the CTG assay (Figure 2B). The Sis for obatoclax, emetine, niclosamide and ganciclovir were 12, 37, 3 and >750, respectively (Appendix A). We titrated HSV-2 produced from drug-treated and non-treated cells in A540-Npro cells (Figure 2C). The experiment revealed that 0.12 μM obatoclax, 0.04 μM emetine, 0.37 μM niclosamide and 0.37 μM ganciclovir lowered the production of HSV-2 in RPE cells. Thus, we identified novel anti-HSV-2 activities of obatoclax, emetine and niclosamide and confirmed the known anti-HSV-2 activity of ganciclovir.

### 3.3. Obatoclax, Emetine, and Homoharringtonin Inhibit EV1 Replication in RPE Cells

Similarly, we examined 43 BSAAs against wild-type EV1 in RPE cells. We monitored the viability of mock- and virus-infected cells by microscopy and the CTG assay. After the initial screening, we identified three compounds (obatoclax, emetine, and homoharringtonine), which rescued cells from virus-mediated death at none-cytotoxic concentrations. To determine the efficiency and toxicity of novel EV1 inhibitors, we measured the viability of mock- and virus-infected cells after 48 h using the CTG assay (Figure 3A). The Sis for obatoclax, emetine, and homoharringtonine were 25, >300, and >300, respectively (Appendix A). We titrated EV1 produced from drug-treated and non-treated cells in A549-Npro cells (Figure 3B). The experiment revealed that all three compounds lowered the production of EV1, confirming novel antiviral activities of obatoclax, emetine, and homoharringtonine.

### 3.4. Brequinar and Suramin Inhibit HIV-1-Mediated Luciferase Expression in TZM-bl Cells

We also examined the toxicity and antiviral activity of 43 BSAAs against HIV-1-mediated firefly luciferase expression. The firefly luciferase open reading frame is integrated into the genome of TZM-bl cells under the HIV-1 LTR promoter. Our primary screen identified two compounds (brequinar and suramin) that suppressed HIV-1-mediated firefly luciferase expression without detectable cytotoxicity. We performed a validation experiment with anti-HIV-1 compounds and calculated selectivity. The SI for brequinar was >750, whereas the SI for suramin was >375 (Figure 4; Appendix A). Thus, we identified novel activity of brequinar and confirmed known activity of suramin.

### 3.5. Obatoclax and Emetine Inhibit RVFV-Mediated RFP Expression in RPE Cells

In addition, we examined the toxicity and antiviral activity of 43 BSAAs against RFP-expressing RVFV in RPE cells. Both fluorescent microscopy and the CTG assay showed that obatoclax and emetine inhibited RVFV-mediated RFP expression at non-cytotoxic concentrations (Figure 5A–C). The SI for obatoclax was >100, and the SI for emetine was >75 (Appendix A).

### 3.6. Obatoclax and Emetine Inhibit HMPV-Mediated GFP Expression in RPE Cells

Next, we tested 43 BSAAs against GFP-expressing HMPV. HMPV-mediated GFP expression and cell viability were measured after 72 h. After the initial screening, we identified two compounds, obatoclax and emetine, which lowered GFP expression without detectable cytotoxicity. Replicate experiments confirmed these hits (Figure 6A–C). The SI for obatoclax was six and for emetine was 10 (Appendix A).

### 3.7. Obatoclax and Emetine Inhibit FLUAV-Mediated GFP Expression in RPE Cells

Next, we tested 43 BSAAs against GFP-expressing FLUAV in RPE cells. After the initial screening, we identified two compounds, obatoclax and emetine, which lowered GFP expression without detectable cytotoxicity. We repeated the experiment with these compounds and confirmed initial hits (Figure 7A–C). The SI for obatoclax was 31 and for emetine was >300 (Appendix A). Thus, we identified novel anti-FLUAV activity of emetine and confirmed known activity of obatoclax.

To test whether the obatoclax-emetine combinations act synergistically against FLUAV-mediated GFP expression and rescue infected cells from death, the observed responses were compared with expected combination responses. The deviations in observed and expected responses showed no synergistic effect (Figure 7D,E; Appendix A). This result indicates that obatoclax and emetine target distinct cellular pathways essential for virus infection.

## 4. Discussion

Here, we tested 43 safe-in-man BSAAs against (−)ssRNA, (+)ssRNA, RT-ssRNA and dsDNA viruses and identified novel activities for five agents (Table 1, Appendix A). We identified novel activities of niclosamide against HSV-2, brequinar against HIV-1, homoharringtonine against EV1, obatoclax against HSV-2, EV1, HMPV and RVFV, and emetine against HSV-2, EV1, HMPV, RVF and FLUAV. We also confirmed antiviral activities of ganciclovir against HSV-2, suramin against HIV-1, and obatoclax against FLUAV [24,30,31]. Our results pointed out that an evasion mechanism observed in one virus could be relevant for other viruses and that existing BSAAs could be re-positioned to other viral infections.

Obatoclax was originally developed as an anticancer agent. Several Phase II clinical trials were completed that investigated the use of obatoclax in the treatment of leukemia, lymphoma, myelofibrosis, and mastocytosis. In addition, its antiviral activity was reported against FLUAV, ZIKV, WNV, YFV, SINV, JUNV, LASV, and LCMV in vitro [24,26,32,33]. It was shown that obatoclax inhibited viral endocytic uptake by targeting cellular induced myeloid leukemia cell differentiation protein Mcl-1 [24]. Given that obatoclax also inhibits RVFV, EV1, HMPV and HSV-2, it could be pursued as a potential BSAA candidate.

Emetine is an anti-protozoal drug. It is also used to induce vomiting. In addition, it possesses antiviral effects against ZIKV, EBOV, RABV, CMV, HCoV-OC43 and HIV-1 [34,35,36,37,38]. It was proposed that emetine can directly inhibit viral polymerases, though it may have some other targets as well [39]. Given that emetine also inhibits FLUAV, RVFV, EV1, HMPV and HSV-2, it may represent a promising BSAA candidate.

Niclosamide is an orally bioavailable anthelmintic drug and potential antineoplastic agent. In addition, it inhibits the broadest range of viruses, including HSV-2, in vitro and, in some cases, in vivo [40,41,42,43,44,45,46,47,48,49]. It was shown that niclosamide induces endosomal neutralization and prevents virus entry into host cells. This supports the further development of niclosamide as a BSAA. 

Homoharringtonine is an anticancer drug which is indicated for treatment of chronic myeloid leukemia. It also possesses antiviral activities against HBV, MERS-CoV, HSV-1 and VZV [50,51,52,53]. Homoharringtonine binds to the 80S ribosome and inhibits viral protein synthesis by interfering with chain elongation [51]. Given that homoharringtonine also inhibits EV1, it may represent a promising BSAA candidate.

Brequinar is an investigational anticancer agent (phase I/II). Brequinar attenuates the replication of DENV, WNV, YFV, LASV, JUNV, LCMV, VSV, HIV-1, and POWV (NCT03760666) [32,54]. It inhibits dihydroorotate dehydrogenase, thereby blocking de novo pyrimidine biosynthesis, which is essential for the transcription and replication of viral RNA. Given that brequinar also inhibits HIV-1, it may represent a promising BSAA candidate.

The human non-malignant RPE cell line represents an excellent model system for studying the infection of different viruses [17,18,24,26]. In addition, different viral strains expressing reporter proteins, such as RVFV-RFP, HMPV-GFP and FLUAV-GFP, are excellent tools for drug screening [17,18,24,27]. However, the number of novel and confirmed antiviral activities of BSAs could be higher if we had used other cell lines and viral strains, different virus loads, different measurement endpoints, a different time of compound addition, as well as a higher purity and concentration range of 43 BSAAs. Moreover, antiviral properties of BSAAs detected in cell-line-based assays might not be reproduced in vivo because systemic mechanisms may compensate the blocked target effect. Thus, follow-up studies are needed to validate our initial hits.

Altogether, we expanded the spectrum of antiviral actions of niclosamide, brequinar, homoharringtonine, obatoclax and emetine in vitro. Importantly, PK and safety studies have been performed on these compounds in laboratory animals and humans. This information could be used to initiate efficacy studies in vivo, saving time and resources. The most effective and tolerable BSAAs or their combinations will have a global impact, improving the preparedness and protection of the general population from emerging and re-emerging viral threats and the rapid management of drug-resistant strains, as well as being used for first-line treatment or for prophylaxis of viral co-infections.

## 5. Conclusions

BSAAs could have a pivotal role in the battle against emerging and re-emerging viral diseases. The development of novel BSAAs could save time and resources which are required for the development of their alternatives—virus-specific drugs and vaccines. In future, BSAAs could have a global impact by decreasing morbidity and mortality from viral and other diseases, maximizing the number of healthy life years, improving the quality of life of infected patients, and decreasing the costs of patient care.

## Figures and Tables

**Figure 1 viruses-11-00964-f001:**
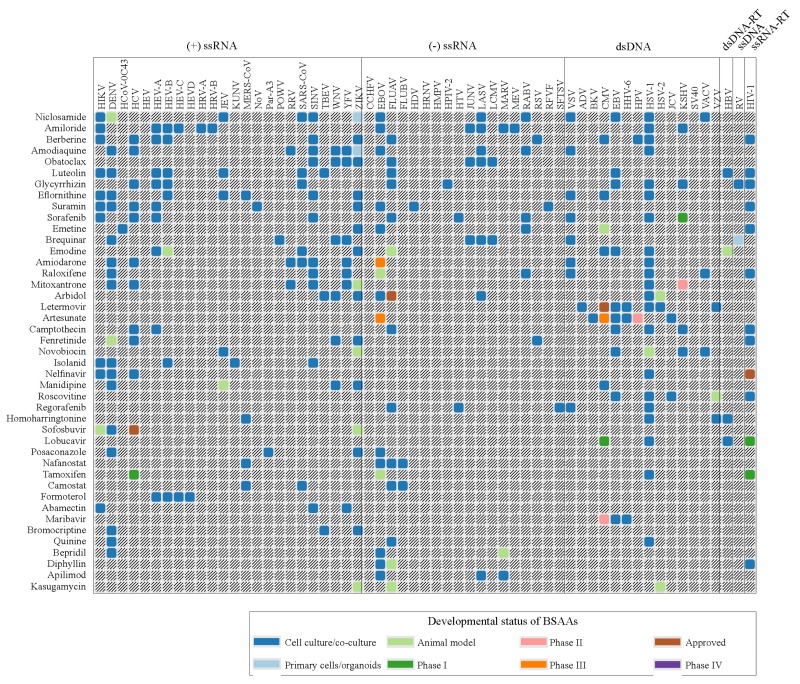
Forty-three safe-in-man broad-spectrum antiviral agents (BSAAs) and viruses that they inhibit. Viruses are clustered by virus groups. BSAAs are ranged from the highest to lowest number of targeted viruses. Different shadings indicate the different development status of BSAAs. Gray shading indicates that the antiviral activity has not been either studied or reported. Abbreviations: ds, double-stranded; RT, reverse transcriptase; ss, single-stranded.

**Figure 2 viruses-11-00964-f002:**
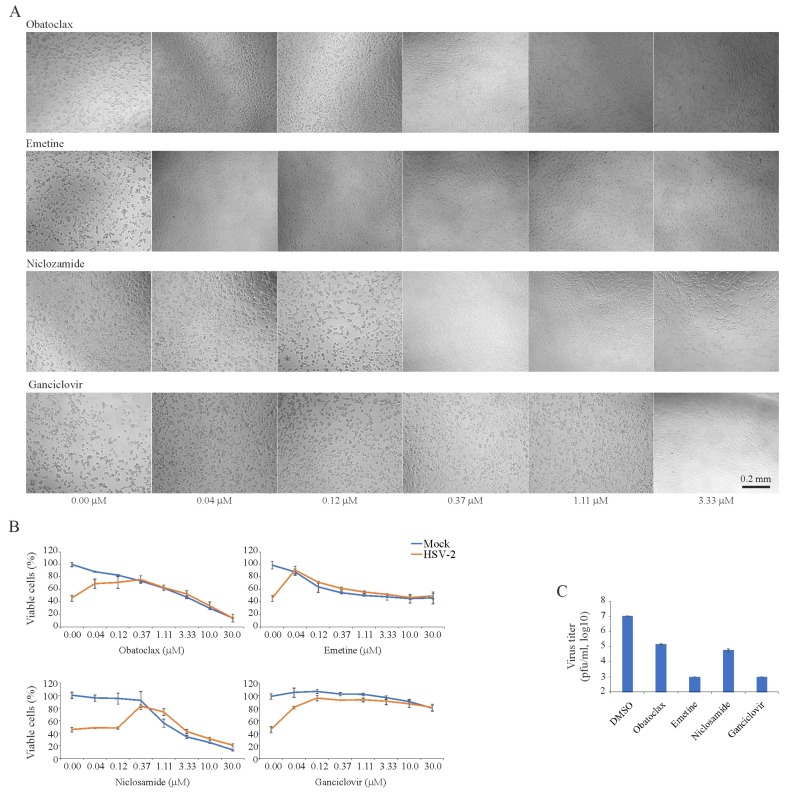
Effect of obatoclax, emetine, niclosamide and ganciclovir on the viability of mock- and herpes simplex virus type 2 (HSV-2)-infected retinal pigment epithelial (RPE) cells and the production of infectious virus particles in these cells. (**A**) RPE cells were treated with increasing concentrations of a compound and infected with HSV-2 (moi, 0.1). Cells were imaged after 72 h in the phase-contrast mode. (**B**) RPE cells were treated with increasing concentrations of a compound and infected with mock- or HSV-2 (moi, 0.1). The viability of the cells was determined to be 72 hpi with the CTG assay. Mean ± standard deviation (SD); *n* = 3 (experimental replicates). (**C**) HSV-2 production in RPE cells treated with obatoclax (0.12 μM), emetine (0.04 μM), niclosamide (0.37 μM), ganciclovir (3.33 μM) or DMSO-treated as measured by plaque assay using A549-Npro cells (Mean ± SD; *n* = 3).

**Figure 3 viruses-11-00964-f003:**
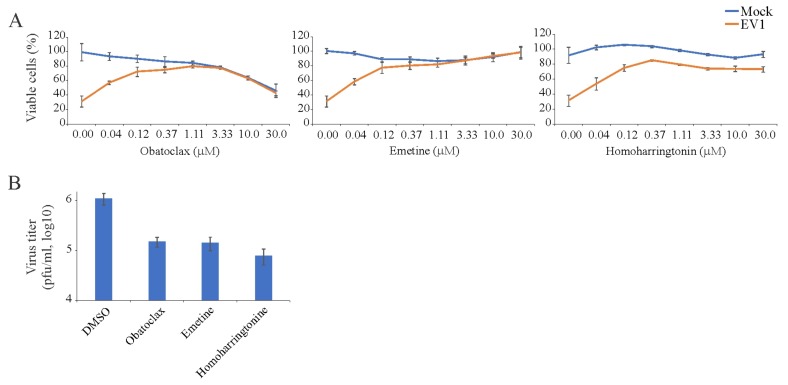
Effect of obatoclax, emetine and homoharringtonine on the viability of mock- and echovirus 1 (EV1)-infected RPE cells and the production of infectious viral particles in these cells. (**A**) RPE cells were treated with increasing concentrations of a compound and infected with EV1 (moi, 0.1). The viability of the cells was determined to be 48 hpi with the CTG assay. (Mean ± SD; *n* = 3). (**B**) EV1 production in RPE cells treated with obatoclax (0.37 μM), emetine (0.12 μM), homoharringtonine (0.12 μM) or dimethyl sulfoxide (DMSO) as measured by plaque assay using A549-Npro cells (Mean ± SD; *n* = 3).

**Figure 4 viruses-11-00964-f004:**
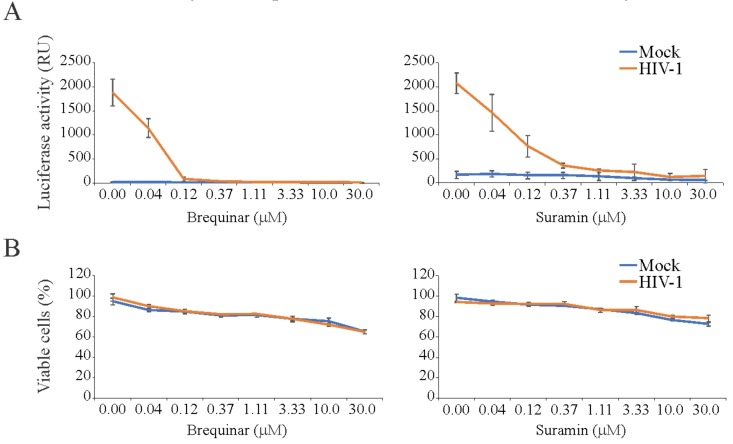
Effect of brequinar and suramin on human immunodeficiency (HIV-1)-mediated luciferase expression and the viability of mock- and virus-infected TZM-bl cells. (**A**) TZM-bl cells were treated with increasing concentrations of a compound and infected with mock or HIV-1 (300 ng/mL of HIV-1 p24). After 24 hpi, the medium was removed from the cells, the cells were lysed, and firefly luciferase activity was measured using the Luciferase Assay System (Mean ± SD; *n* = 3). (**B**) In a parallel experiment, Cell Tox Green reagent (CTxG) reagent was added to the cells and the fluorescence was measured. The percentage of viable cells was calculated. Mean ± standard deviation (SD); *n* = 3 (experimental replicates).

**Figure 5 viruses-11-00964-f005:**
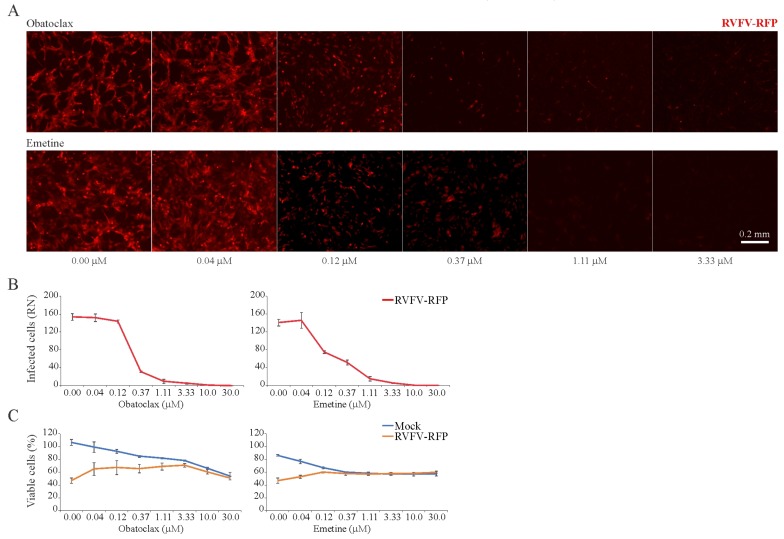
Effect of obatoclax and emetine on Rift Valley fever virus (RVFV)-mediated red fluorescent protein (RFP) expression and on the viability of mock- and virus-infected RPE cells. (**A**,**B**) RPE cells were treated with increasing concentrations of a compound and infected with RVFV-RFP (moi, 1). After 24 h, the cells were imaged and fluorescence intensity was quantified (Mean ± SD; *n* = 3). (**C**) Cell viability was determined using the CTG assay (Mean ± SD; *n* = 3).

**Figure 6 viruses-11-00964-f006:**
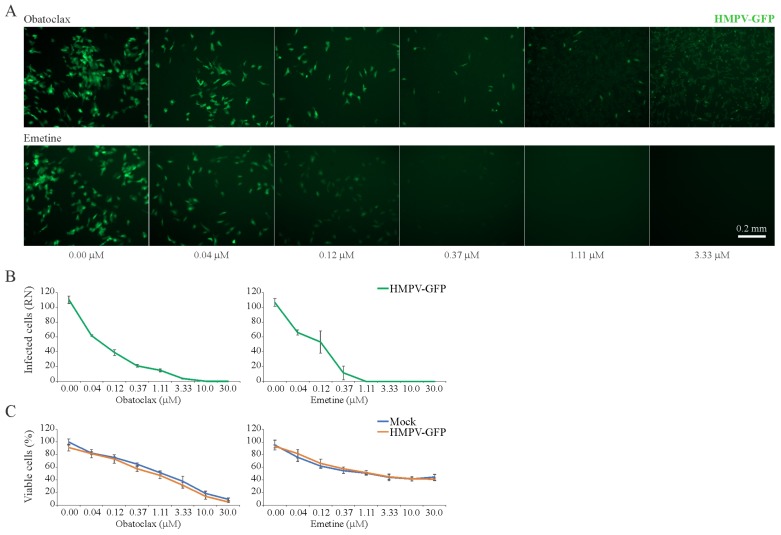
Effect of obatoclax and emetine on metapneumovirus (HMPV)-mediated green fluorescent protein (GFP) expression and on the viability of mock- and virus-infected RPE cells. (**A**,**B**) RPE cells were treated with increasing concentrations of a compound and infected with HMPV-GFP (moi, 0.1). After 96 h, the cells were imaged and fluorescence intensity was quantified (Mean ± SD; *n* = 3). (**C**) Cell viability was determined with the CTG assay (Mean ± SD; *n* = 3).

**Figure 7 viruses-11-00964-f007:**
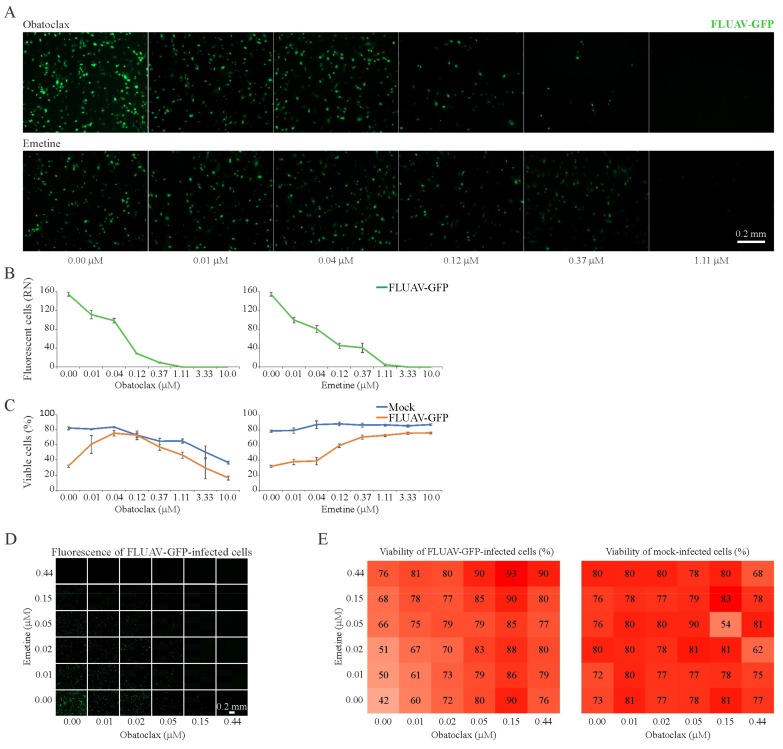
Effect of obatoclax, emetine and their combinations on influenza A virus (FLUAV)-mediated GFP expression and the viability of mock- and virus-infected RPE cells. (**A**,**B**) RPE cells were treated with increasing concentrations of a compound and infected with FLUAV-GFP (moi, 0.5). After 24 h, the cells were imaged and fluorescence intensity was quantified (Mean ± SD; *n* = 3). (**C**) Cell viability was determined with the CTG assay (Mean ± SD; *n* = 3). (**D**–**E**) The interaction landscapes of the emetine-obatoclax drug combination on FLUAV-mediated GFP expression and the viability of FLUAV- and mock-infected RPE cells, as measured with CTG assay.

**Table 1 viruses-11-00964-t001:** The half-maximal cytotoxic concentration (CC_50_), the half-maximal effective concentration 1 (EC_50_-1) calculated using the CTG or CTxG assays, EC_50_-2 calculated using reporter protein expression, and the minimal selectivity indexes (SIs = CC_50_/EC_50_) for selected broad-spectrum antivirals. The measurements were repeated three times.

Compound	Virus	Cell Line	Time (h)	CC_50_ (mM)	EC_50_-1 (mM)	EC_50_-2 (mM)	SI
Obatoclax	HSV-2	RPE	72	1.23 ± 0.02	0.10 ± 0.02		12
Emetine	HSV-2	RPE	72	1.12 ± 0.07	0.03 ± 0.01		37
Niclosamide	HSV-2	RPE	72	1.31 ± 0.02	0.43 ± 0.04		3
Ganciclovir	HSV-2	RPE	72	>30	0.04±0.01		>750
Obatoclax	EV1	RPE	48	3.21 ± 0.04	0.12 ± 0.01		25
Emetine	EV1	RPE	48	>30	0.12 ± 0.04		>300
Homoharringtonine	EV1	RPE	48	>30	0.14 ± 0.03		>300
Brequinar	HIV-1	TZM-bl	24	>30		0.04 ± 0.01	>750
Suramin	HIV-1	TZM-bl	24	>30		0.08 ± 0.03	>375
Obatoclax	RVFV	RPE	24	>30	0.04 ± 0.01	0.32 ± 0.09	>100
Emetine	RVFV	RPE	24	>30	0.10 ± 0.02	0.43 ± 0.10	>75
Obatoclax	HMPV	RPE	96	0.60 ± 0.04		0.12 ± 0.02	6
Emetine	HMPV	RPE	96	1		0.14 ± 0.05	10
Obatoclax	FLUAV	RPE	24	3.11 ± 0.09	0.04 ± 0.01	0.10 ± 0.01	31
Emetine	FLUAV	RPE	24	>30	0.13 ± 0.05	0.12 ± 0.03	>300

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
