# Peer review of "Novel Antiviral Activities of Obatoclax, Emetine, Niclosamide, Brequinar, and Homoharringtonine"

_viruses, 2019, doi:10.3390/v11100964_

Round 1

Reviewer 1 Report

Please find attached this reviewer's comments.

Reviewer 2 Report

The authors did an important effort to discover and characterise broad-spectrum antivirals. However we believe the current state of the research does not meet the general standards in the field.

Major comments:

The authors determined EC50s and CC50s for several virus/antiviral combinations but most are single measurements. The standard of reporting antiviral activities involves multiple experiments that yield independent measurements of EC50s and CC50s and that allows some statistical analysis (for proper summarization of EC50s see

Reviewer 3 Report

1. Drug repurposing (also called repositioning, redirecting, reprofiling) is a polypharmacology-driven
strategy for generating additional value from an existing drug by targeting diseases other than that for
which it was originally intended. This has significant advantages over new drug discovery since
chemical synthesis steps, manufacturing processes, reliable safety, and pharmacokinetic properties in
early clinical developmental phases (Phase I and Phase IIa trials) are already available.
Therefore, repositioning of launched or even failed drugs provides unique translational opportunities,
including a substantially higher probability of success to market as compared with new drugs, and a
significantly reduced cost and timeline to clinical availability.
More recent references on drug repositioning should be cited.
2. Emetine was once reported as an HIV inhibitor, thus the following reference should be cited:
Natural Plant Alkaloid (Emetine) Inhibits HIV-1 Replication by Interfering with Reverse Transcriptase
Activity. Chaves Valadão AL, Abreu CM, Dias JZ, Arantes P, Verli H, Tanuri A, de Aguiar RS. Molecules.
2015 Jun 22;20(6):11474-89.
3. In the biological evaluation, the control drugs were missing. The authors should select the
approved drugs or drug candidates as controls.

Author Response

Drug repurposing (also called repositioning, redirecting, reprofiling) is a polypharmacology-driven strategy for generating additional value from an existing drug by targeting diseases other than that for which it was originally intended. This has significant advantages over new drug discovery since chemical synthesis steps, manufacturing processes, reliable safety, and pharmacokinetic properties in early clinical developmental phases (Phase I and Phase IIa trials) are already available. Therefore, repositioning of launched or even failed drugs provides unique translational opportunities, including a substantially higher probability of success to market as compared with new drugs, and a significantly reduced cost and timeline to clinical availability. More recent references on drug repositioning should be cited.

Re 3.1: Many thanks for positive feedback. We now added 3 additional references to drug repositioning approaches (PMID: 30310233, 30294274 and 30123072): “In order to overcome these time and cost issues, academic institutions and pharmaceutical companies focused on repositioning of existing antivirals from one viral disease to another, taking into account that the most viruses utilize the same host factors and pathways to replicate inside a cell [6-15].”

Emetine was once reported as an HIV inhibitor, thus the following reference should be cited: Natural Plant Alkaloid (Emetine) Inhibits HIV-1 Replication by Interfering with Reverse Transcriptase Activity. Chaves Valadão AL, Abreu CM, Dias JZ, Arantes P, Verli H, Tanuri A, de Aguiar RS. Molecules. 2015 Jun 22;20(6):11474-89.

Re 3.2: We have the reference to emetine as anti-HIV-1 agent in Supplementary Table 1 (PMID: 26111177).

In the biological evaluation, the control drugs were missing. The authors should select the approved drugs or drug candidates as controls.

Re 3.3: We used broad-spectrum antiviral agents, saliphenylhalamide and ABT-263, as controls in our initial screens. Saliphenylhalamide is inhibitor of cellular vacuolar ATPase, which protects cells from virus-mediated death, whereas ABT-263 is inhibitor of anti-apoptotic Bcl-2 proteins, which facilitates death of infected cells in cooperation with viral nucleic acids. We now added the sentence in text: “Saliphenylhalamide, an inhibitor of viral endocytosis, ABT-263, a stimulator of infected cell death, or DMSO were added to the control wells [16-21].”

Round 2

Reviewer 2 Report

The authors have now included information on the reproducibility in Table S4. In our opinion this table is of great interest to the reader and would better be taken up in the manuscript (example discussion). Unfortunately Table S4 is not well described in the manuscript or in the table legend (does it come from independent experiments? How many repeats?). Based on the fact that there is information on p-values and on the rebuttal of the authors I believe TableS4 represents the summary of 3 independent experiments but please add this information in the manuscript. The authors should also include the variation on EC50 or CC50 (interquartile range? standard deviation?) in the table and perhaps also when they mention these values in the text. It is a good idea to show a p-value but please explain how it was calculated  (unpaired t-test?).

Figure 1 and Figure 8 contain the same information. I prefer Fig 1 as from Fig 8 one cannot always see where the lines start and stop. On Fig 1 the authors can also indicate the new interactions they have discovered.

The authors should thoroughly go through the text for typos like:

“… considering that the most viruses …” “… thus we expanded spectrum of activities …”

Author Response

Many thanks for your suggestions. As suggested, we moved Table S4 to discussion section (now it's Table 1) and Fig. 8 to supplementary section (now it's Fig. S2). We also included SD to CC50 and EC50 values in Table S1 and stated that the experiments were repeated 3 times. In addition, we went thoroughly through the text and corrected typos like: “… considering that the most viruses …” to “… considering that many viruses …”. The “… thus we expanded spectrum of activities …”  was omitted.